# Modeling Human Eye Movements with Neural Networks in a Maze-Solving Task

**Jason Li**                                                    jasli@mit.edu
**Nicholas Watters**                                          nwatters@mit.edu
**Yingting Wang**                                            swang22@bu.edu
**Hansem Sohn**                                              hansem@mit.edu
**Mehrdad Jazayeri**                                          mjaz@mit.edu
*Department of Brain and Cognitive Sciences,*
*McGovern Institute for Brain Research*
*Massachusetts Institute of Technology*
*Cambridge, MA 02139*

## Abstract

From smoothly pursuing moving objects to rapidly shifting gazes during visual search, humans employ a wide variety of eye movement strategies in different contexts. While eye movements provide a rich window into mental processes, building generative models of eye movements is notoriously difficult, and to date the computational objectives guiding eye movements remain largely a mystery. In this work, we tackled these problems in the context of a canonical spatial planning task, maze-solving. We collected eye movement data from human subjects and built deep generative models of eye movements using a novel differentiable architecture for gaze fixations and gaze shifts. We found that human eye movements are best predicted by a model that is optimized not to perform the task as efficiently as possible but instead to run an internal simulation of an object traversing the maze. This not only provides a generative model of eye movements in this task but also suggests a computational theory for how humans solve the task, namely that humans use mental simulation.

**Keywords:** recurrent neural network, saccade, mental simulation, psychophysics, maze, gaze, fovea

## 1. Introduction

Throughout the history of cognitive science, eye movements have been appreciated as a window into the workings of the mind and brain (Helmholtz, 1924; Liversedge and Findlay, 2000; Hayhoe and Ballard, 2005; König et al., 2016). However, human eye movements are so rich and varied that characterizing them is difficult even in simple tasks (Land and McLeod, 2000; Beller. et al., 2022; Gerstenberg et al., 2017). Building generative models of eye movements is an even greater challenge (Chen et al., 2017; Zoran et al., 2020), and to date most such work focuses only on free-viewing or visual search contexts, not complex cognitive tasks (Kümmerer and Bethge, 2021; Zelinsky et al., 2020).

To tackle the problem of modeling task-driven saccade sequences, we designed a maze-solving task. In this task, subjects must find the exit location of a path in a maze given a starting point of the path (Figure 1). This task provides an ideal platform for building generative models of eye movements because it offers a near-limitless variety of spatial plans,

yet eye movements are largely consistent across humans (Crowe et al., 2000), making them tractable to model. Furthermore, this task may be solved using mental simulation of an object traveling through the maze, so allows us to test mental simulation as a computational theory guiding eye movements (Gerstenberg et al., 2017; Ullman et al., 2017; Ahuja and Sheinberg, 2019; Rajalingham et al., 2021).

In this work, we develop a novel general-purpose method for incorporating features of human vision such as eccentricity-dependent visual acuity and discrete saccades into a task-optimized, end-to-end differentiable recurrent network. Using this method, we construct a space of models with and without mental simulation constraints, and train these on the maze-solving task. We collect eye movement data from human subjects playing the task and compare this data to eye movements generated by the models to test multiple hypotheses for how humans solve the task.

## 2. Related Work

Building generative models of human eye movements has been an active area of research in psychology for decades (Zelinsky et al., 2020; Kümmerer and Bethge, 2021; Wedel et al., 2022). One approach to modeling eye movements is to hard-code the heuristics of eye movements without employing task-driven learning. This approach has seen some success in free viewing or visual search contexts (Itti et al., 1998; Zelinsky, 2008; Zhang et al., 2005; Adeli et al., 2017; Zelinsky et al., 2013; Eckstein, 2011). However, our model differs from those approaches in that (i) it learns a general policy for generating saccadic eye movements, so can in principle be applied to any task, and (ii) is a neural network, hence can more easily serve as a mechanistic model of the brain at an implementation level.

More recently, deep learning approaches to generate sequences of eye movements have been developed, for example, by fitting a model directly to human data (Assens Reina et al., 2017; Sun et al., 2019; Xia et al., 2019; Yang et al., 2020; Kümmerer et al., 2022). While these can provide impressive fits to human data, in this work our goal differs in that we aim to build a model that shows emergent human-like eye movements through task optimization, without explicitly fitting to human eye movement data. Other deep learning models employ sequential attention in general-purpose task-trained networks, but have not been hypothesized as models of human eye movements or tested against human eye data (Gregor et al., 2015; Eslami et al., 2016; Adeli et al., 2022). In contrast, we develop our models to generate human-like eye movements, collect (and open-source) human eye data, and test our models against this data.

## 3. Methods

All of our data and code can be found at https://github.com/jazlab/Maze_Task_2022, along with documentation and instructions for replicating our results.

### 3.1. Task and maze dataset

In the maze-solving task, a subject is presented with a square maze and an entrance point somewhere on its perimeter. This entrance point is one end of an unbroken, non-branching path through the maze, which exits the maze at some other uniformly sampled perimeter

point. The subject is tasked to find this exit point (see Appendix A for task instructions for human subjects). We trained models on random mazes generated online. We also created a test set for human and model comparison comprising 200 unique procedurally generated mazes. See Appendix B for details about the maze generation algorithm.

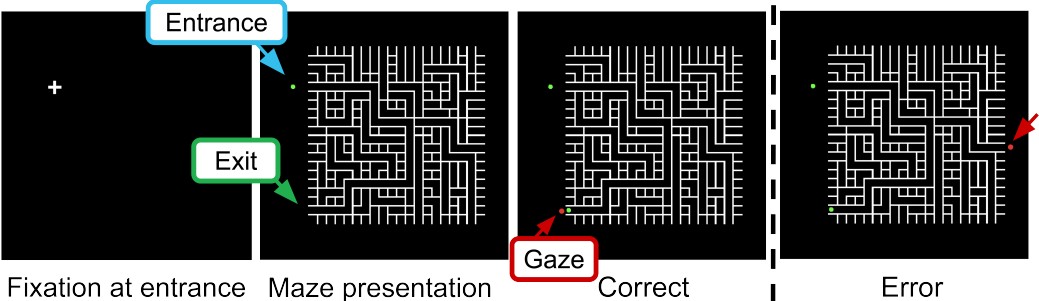

Figure 1: Screenshots of the task presented to human subjects. At the start of a trial, the subject fixates at a maze entrance point indicated by a white cross (before maze presentation) or a green dot (after maze presentation). Subjects must locate the correct exit point and press a button once they have fixated their eye gaze on the exit point. At the button press, the true exit position is indicated by another green dot (lower-left side in "Correct") and a red dot ("Gaze") shows the reported position. The rightmost panel ("Error") illustrates an error trial when subjects incorrectly identified the exit.

### 3.2. Human data collection

Fourteen human subjects volunteered to participate in the experiments after providing informed consent. All participants (age: 18 to 65 years old, eight female and six male) had normal or corrected-to-normal vision with no history of neurological or psychiatric disorders. All experiments were approved by the Committee on the Use of Humans as Experimental Subjects at the Massachusetts Institute of Technology.

Subjects were seated in front of a LCD monitor (width: 53 cm, height: 30 cm; Acer R240HY) at a distance of 66 cm. Each session started with a procedure for calibrating eye positions using an optical eye tracker (EyeLink 1000 Plus, SR Research). Eye position was monocularly sampled with 1 ms resolution while a chin rest stabilized head of the participants. We monitored quality of the eye signal during the experiment and repeated the eye calibration in the middle of the session if needed. After receiving instruction for the maze-solving task and several practice trials, data collection began. In each trial, we randomly selected a maze from a predetermined test set. The test set included two repetitions of each unique maze. Each participants completed one 1-hour session, which consisted of approximately 400 trials. Maze stimuli (14 degree of visual angle) and behavioral contingencies were controlled by an open-source software (MWorks; mworks-project.org/) and Modular Object-Oriented Games (MOOG) library (Watters et al., 2021).

While solving the maze, humans' eye movements were exclusively saccadic. Consequently, we extracted saccades from the eye position data by first filtering with a 4 ms

Gaussian kernel and then thresholding eye velocity at 50 degrees of the visual angle per second. To prevent measurement noise of the eye position from dominating metrics that we use to compare humans with models, we recalibrated the raw eye position data for each trial. To do so, we estimated a calibration error vector between the actual fixation point and the gaze fixation point, and subtracted that vector from eye positions throughout the trial. This recalibration did not affect our main findings.

### 3.3. Gaze recurrent neural network (RNN) models

We developed a task-optimized recurrent convolutional neural network model that is equipped with a foveal module. The foveal module allows the model to receive high acuity visual information near the fovea and low acuity information in the periphery, like the human eye. The recurrent model is also able to control the position of its fovea, allowing it to make eye movements. The model is end-to-end differentiable, so can be trained via backpropagation, through which an eye movement policy emerges from task-optimization.

We modeled the fovea by applying a circular exponential mask $e^{-d/\tau}$ to the visual input, where $d$ is distance to the center of fovea and $\tau$ is a scaling parameter. We chose 5 pixels as a value of $\tau$ (within a maze of 39 pixels), which is consistent with reported human peripheral visibility maps (Najemnik and Geisler, 2005; Strasburger et al., 2011). See Appendix C for results with varying choices of $\tau$.

After applying the mask, we add independent noise to each pixel, sampled from a normal distribution $\mathcal{N}(\mu = 0, \sigma^2 = 0.05)$. This noise washes out faint information in the peripheral tail of the foveal mask, analogous to the decreased peripheral photoreceptor density in the human retina. Note that this noise is essential to prevent the network from exploiting the peripheral information available when only the mask is used. Figure 2(a) shows a diagram of the model's fovea mechanism. This foveal module is general-purpose, and in theory can be incorporated into any RNN that takes visual input. To our knowledge, this method is novel in the field.

We implement three specific convolutional RNN models, EXIT, SIM, and HYBRID. They all receive visual input via this foveal module, have the same internal architecture, and can generate two outputs, the Cartesian coordinate for the eye position (i.e., the center of fovea) and the Cartesian coordinate of the next ball position (Figure 2(b)). The three models differ only in their objective function.

### 3.3.1. EXIT

The EXIT model is trained with a Mean Squared Error (MSE) loss between the eye position at each step ($\hat{p}_i^{\text{eye}}$) and the maze exit point ($p^{\text{exit}}$), across all $n$ steps. There is no loss on the model's ball position output. This model represents an optimal exit-finding strategy where the model moves its eyes to the exit in as few saccades as possible. We minimize

$$L_{\text{EXIT}} = \frac{1}{n} \sum_{i=1}^{n} (\hat{p}_i^{\text{eye}} - p^{\text{exit}})^2$$

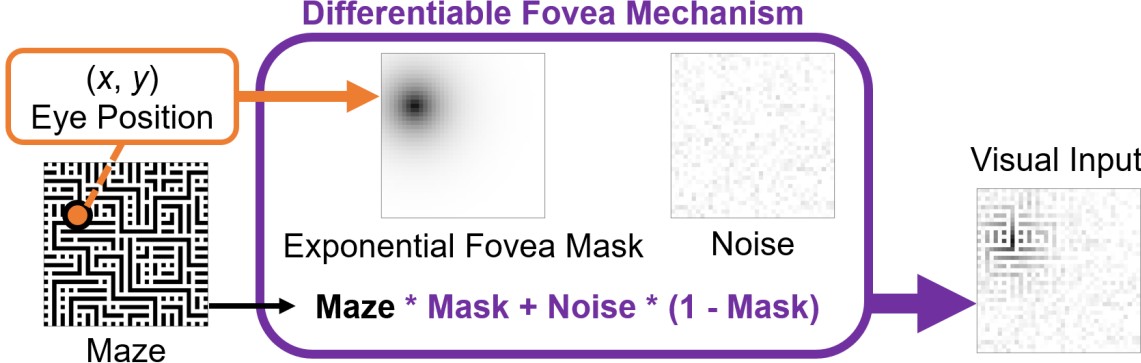

(*a*) Differentiable fovea mechanism. Input is eye position and rendered maze; output is noisy masked maze, the visual input for the next step of the gaze RNN.

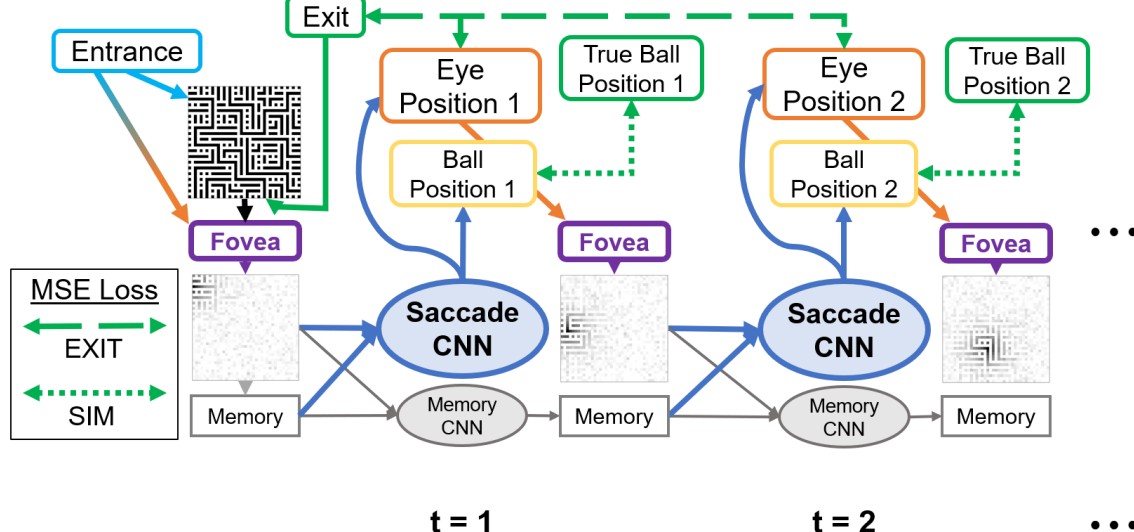

(*b*) Gaze RNN model. Memory CNN is a 3-layer Convolutional Neural Network. Saccade CNN is a strided 3-layer CNN with two 3-layer MLP (Multi-Layer Perceptron) heads for Cartesian eye position and ball position vectors. This architecture can be unrolled through time for an arbitrary number of steps or saccades.

Figure 2: Gaze RNN model diagram.

### 3.3.2. SIMULATION

The SIM model aims to predict the position of an imaginary ball that moves from start to exit at constant velocity. To do so, we optimize this model with a "simulation loss," which is formulated as the MSE between each predicted ball position ($\hat{p}_i^{\text{ball}}$) and the actual position of an imaginary ball traveling at 10 pixels per time step ($p_i^{\text{ball}}$). This ball speed was calculated from the human eye movement data as $l/n$ averaged over all trials, where $l$ is total maze length and $n$ is number of saccades. In the SIM model, eye position is not

explicitly constrained, but still plays a critical role in advancing the model's visual field. We minimize

$$L_{\text{SIM}} = \frac{1}{n} \sum_{i=1}^{n} (\hat{p}_i^{\text{ball}} - p_i^{\text{ball}})^2$$

### 3.3.3. HYBRID

The HYBRID model is trained with a weighted sum of the loss functions for the EXIT and SIM models. The model's eye position must reach the exit quickly *and* allow the model's predicted ball positions to match the position of the imaginary ball in the maze. The ratio of SIM to EXIT loss weight is controlled by a coefficient $\beta = \frac{1}{3}$, chosen so that the two loss terms have similar magnitudes in a fully trained model. We minimize

$$L_{\text{HYBRID}} = \beta \cdot L_{\text{EXIT}} + (1 - \beta) \cdot L_{\text{SIM}}$$

All models were trained on an NVIDIA GeForce GTX 1080 TI GPU with 4 GB RAM per model and a total compute time of about 100 hours. Given computational resource limitations, we trained one instance of each model, though see Appendix C for results from additional instances in the context of hyperparameter sweeps. Models were implemented in PyTorch (Paszke et al., 2019) and trained with 8 recurrent steps per maze, batch size 16, through 1.8 million iterations using Adam optimizer (Kingma and Ba, 2014) with learning rate 0.0003. This was sufficient for each model's loss to converge to a stable asymptote. The training dataset was generated online using a custom procedural maze generator with the same statistics (though not the same samples) as the test set (see Appendix B for maze-generation details).

### 3.4. Baseline model

As a standard of comparison for the gaze RNNs, we created a baseline model designed to match high-level human saccade statistics. This model iteratively constructs saccade paths where the amplitude and angle of each saccade, as well as the total number of saccades in the path, are sampled from the corresponding distributions found in our human eye movement data (Figure 5). For each trial, we construct 2,000 saccade paths and select the path whose final fixation point is closest to the correct maze exit. For our maze test set, this is sufficient to guarantee that the final fixation point falls within 5 pixels of the maze exit greater than 90% of the time.

### 3.5. Metrics for comparing eye movement data

To quantify these results, we use two metrics for comparing eye movement paths:

- **Nearest neighbors distance** is computed as the mean of the nearest point in path $A$ to each point $p_B$ in path $B$ and the nearest point in path $B$ to each point $p_A$ in path $A$:

$$\mathcal{L}_{NN} = \frac{1}{2} \cdot \left( \mathbb{E}_{p_A \in A} \left[ \min_{p_B \in B} ||p_B - p_A||_2 \right] + \mathbb{E}_{p_B \in B} \left[ \min_{p_A \in A} ||p_A - p_B||_2 \right] \right)$$

- **Area between paths** is computed as the total area of the polygon(s) formed between paths $A$ and $B$. See Appendix D for details.

For both of these metrics, a lower value implies the paths $A$ and $B$ are more similar.

## 4. Results

### 4.1. Saccade path similarity

Figure 3 shows the behavior of two representative human subjects, three RNN models, and the baseline model for three example mazes in the test set. Evidently, the saccadic eye movements in humans and models roughly follow the correct path through the maze and successfully find the exit point. Humans display a tendency to cut corners of the maze path. Qualitatively, out of the gaze RNNs, the EXIT model seems most dissimilar to humans as it often makes large saccades that are not present in human eye movements. The SIM and HYBRID models make more uniform saccades that appear to better match human saccades. The baseline model tends to generate erratic saccade paths that, by construction, terminate near the exit point but do not resemble human saccade paths.

Quantitatively, the SIM model exhibits the lowest mean model-human distances under both distance metrics (Figure 4). On the other hand, the EXIT model produces the least human-like eye movement paths, comparable to those produced by the baseline model. The HYBRID model's metric scores fall between those of the other two gaze RNN models. Therefore, the SIM model is the most human-like of our three generative models.

### 4.2. Saccade vector similarity

In addition to comparing human and model saccade paths on a trial-by-trial basis, we also compared their aggregate saccade vector distributions. Figure 5 shows a sample of the saccades executed by humans and each model on the test set, with the tail of every saccade vector centered and the head plotted as a point. In the human distribution, saccade angle is relatively uniform and most saccade amplitudes are contained within a radius of 3 degrees. This is also true for the baseline model's distribution, which was sampled from the human distribution. Consistent with the sample shown in Figure 3, the EXIT model's distribution shows many high-amplitude saccades, reflecting the large, erratic saccades that model tends to favor. The SIM model's distribution is nearly bounded by a square, which results from the right-angle maze geometry. Saccade amplitude attains its maximum at the four cardinal directions because it is in those directions that a constant-velocity ball can travel farthest through the maze.

These results suggest that although the SIM model's eye movement paths most closely resemble those of humans, it is not a complete model of humans' eye movement strategy when solving this task.

## 5. Limitations and Future Work

One limitation of our work is that the simulation model shows a more constrained saccade vector distribution than humans (Figure 5), which may be a consequence of its tendency to follow the path more faithfully than humans (Figure 3). This close path-following with

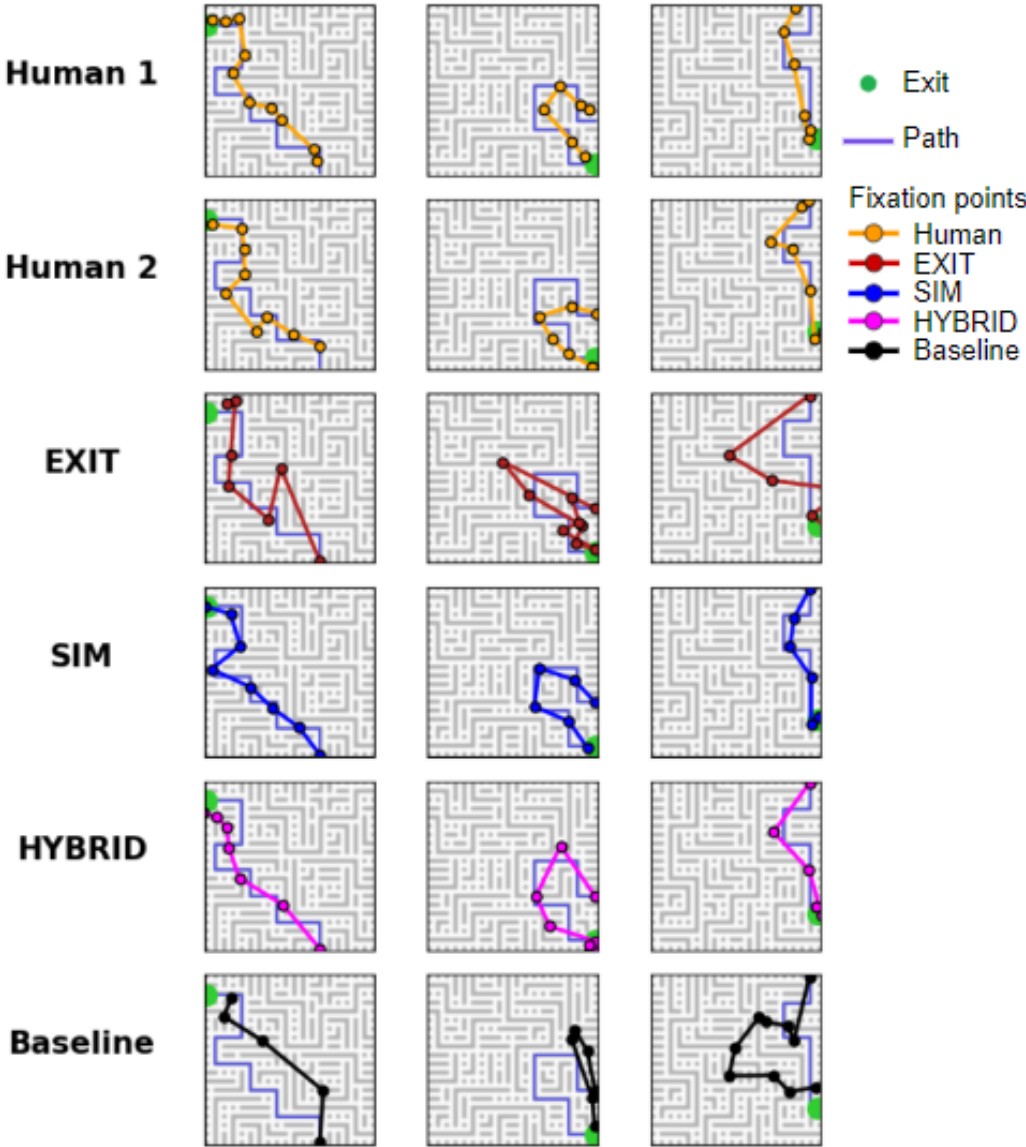

Figure 3: Human and model behaviors on three sample mazes.

its fovea is an emergent property of the model: The simulation loss was applied only to the model's "ball position" output, not its "eye position" output. Nonetheless, similarity to human eye movements may be improved with variants of the SIM model, such as by (i) varying the speed of the ball simulation, or (ii) introduce a non-constant simulation speed, either learned by the model or computed based on predetermined heuristics (such as speeding up on long straightaways and slowing down near corners). Furthermore, exploration of the impact of simulation speed on eye movements in the model may lead to predictions about simulation speed in humans. Future work is needed to explore these possibilities.

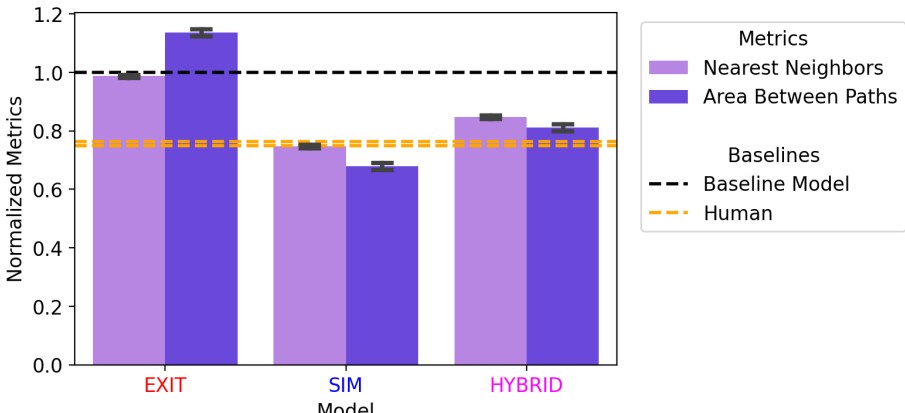

Figure 4: Metric scores between model and human eye paths computed on the test set. To compute these given a model and metric, for each test maze we compute the metric score on each [model gaze path, subject gaze path] pair using all human subject trials and 2 runs of a trained instance of the model on the test maze. We then average all of these scores to obtain a total model-human similarity. Error bars are 95% confidence intervals. Quantitatively, nearest-neighbors and area-between-paths scores are: EXIT: [0.987 ± 0.005, 1.135 ± 0.012]; SIM: [0.747 ± 0.007, 0.679 ± 0.012]; HYBRID: [0.848 ± 0.006, 0.811 ± 0.013]; between-human mean: [0.762, 0.750]. Note that the SIM models achieves better average similarity to humans than between-human similarity, which is not impossible and implies that the model has lower variance than the inter-subject variance.

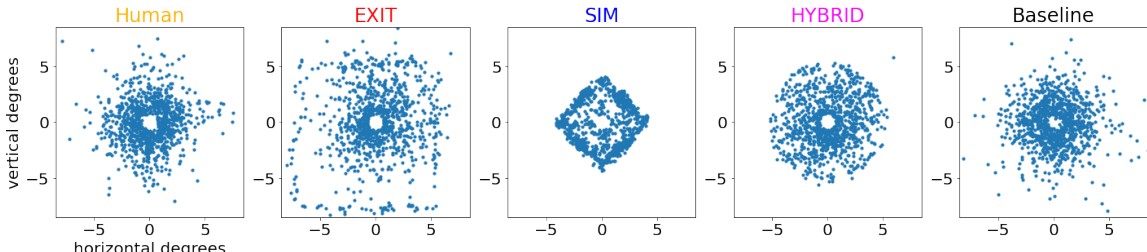

Figure 5: Saccade vector distributions for humans and models in visual angle. For models, each plot shows 1,000 randomly sampled saccade vectors from a dataset of 2 model evaluations on each test maze. The human plot shows 1,000 randomly sampled saccades from all human data aggregated across subjects.

A second limitation of our work is that it is difficult to characterize the biological plausibility of our foveal module's hyperparameters. For instance, the human visibility varies greatly based on context-dependent factors like scene clutter, crowding, and luminance

(Pelli et al., 2007; Levi, 2008; Strasburger et al., 2011), so it is difficult to ascertain how our scaling parameter $\tau$ compares to that of the human fovea. Nevertheless, we believe that our choice of hyperparameters falls within reasonable bounds and when we varied the hyperparameter $\tau$, the simulation model is still the most similar to the human data (Appendix C).

A third limitation of our models is that they make one saccade for each RNN timestep. This prevents them from capturing temporal aspects of human eye movements, such as the duration of fixations between saccades, reaction times, etc. Future work may address this by allowing the model to control fixation durations, which could emerge from task training if longer fixations reduce perceptual noise.

Finally, the mazes used to train and test the gaze RNN models had paths and walls of the same width, while those presented to human subjects playing the task had much thinner walls than paths. The resolution of the gaze RNN mazes was limited by computational capacities. This may have resulted in humans producing slightly different eye movement trajectories.

## 6. Conclusion

We find that in the maze-solving task, a gaze RNN trained to run an internal simulation of a ball moving through a maze generates eye movements more similar to those of human subjects than a model trained only to solve the task as optimally as possible. This suggests that humans may employ a similar mental simulation when performing this maze-solving task. Further work is needed to explore the relationship between the biological plausibility of the model fovea hyparparameters and model behavior. Future work also includes incorporating our differential fovea method into RNNs trained on other tasks to study the principles of human eye movements in domains beyond maze-solving.

## Acknowledgments

J.L. is supported by the MIT Quest for Intelligence. N.W. is supported by the National Science Foundation. Y.W. is supported by the Simons Foundation. H.S. is supported by a NARSAD young investigator grant from the Brain & Behavior Research Foundation. M.J. is supported by the Simons Foundation, the McKnight Foundation, and the McGovern Institute.

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

## Appendix A. Task instructions for human subjects

The goal of this "maze-solving" task is to identify the exit of the maze given an initial entry position (indicated by a green ball). At the beginning of a trial, you will see a white cross, which you are asked to look at. The position of the cross corresponds to the initial entry position.

After a random delay, the maze and the green ball will appear on the screen. From the entry where the green ball sits in, there will be one continuous path until it hits one of the 4 boundaries of the maze.

After you identify the exit, fixate your eye gaze at the exit and then press the left-arrow key to report the exit location (i.e., where you are looking when pressing the key). As a

feedback, the green ball will be revealed at the correct location where it should exit the maze. Your reported exit location will be shown with a red ball. Therefore, if both green and red ball are close or right next to each other, that means your response was correct.

Note that we will record your eye gaze throughout the trial and so do your best not to move your head during the experiment.

Every 50 trials, a gray screen will appear and you are given a break as long as you want. When you are ready to resume the task, press the left-arrow key again. In total, you will complete 400 trials.

## Appendix B. Maze Generation

We procedurally generated maze via a simple layering procedure. First, we implemented a path-generation algorithm, which sampled a random path within the $20 \times 20$ maze grid by picking a random starting edgepoint and taking a random walk with turn probability 0.2 and minimum inter-turn distance 3 until reaching an edgepoint. Then, to generate a maze we layered such randomly generated paths with occlusion until every grid location in the maze was covered by some path.

The code for this algorithm can be found in our open-sourced repo: https://github.com/jazlab/Maze_Task_2022.

## Appendix C. Fovea Size Sweeping

Table 1: Similarity of gaze RNNs to humans across three values of $\tau$.

| (a) Nearest neighbors | | | | | (b) Area between paths | | | |
|---|---|---|---|---|---|---|---|---|
| $\tau$ (pixels) | EXIT | SIM | HYBRID | | $\tau$ (pixels) | EXIT | SIM | HYBRID |
| 8 | 5.04 | 3.10 | 3.71 | | 8 | 294.3 | 134.4 | 202.7 |
| 5 | 4.65 | 3.11 | 3.85 | | 5 | 270.0 | 137.9 | 191.7 |
| 3.33 | 4.36 | 3.14 | 3.15 | | 3.33 | 242.8 | 139.2 | 142.6 |

## Appendix D. Area between paths metric

The area-between-paths metric measures similarity of two paths in space. It is computed as the total plane area of all polygon(s) formed between the two paths. See Figure 6 for an illustration.

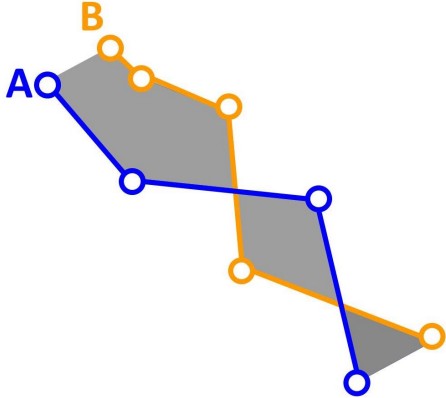

Figure 6: Illustration of area between paths. Given paths A and B, the area between them is shaded gray.

