# OpenReview forum: "Modeling Human Eye Movements with Neural Networks in a Maze-Solving Task"
_NeurIPS.cc/2022/Workshop/GMML — Gaze Meets ML 2022 Oral_

### Official Review · Reviewer_6Q1d · 2022-10-18
**Well designed, well-written study with a not overly complicated model that its rationale is rooted in the human visual system**

**Rating:** 8
**Confidence:** 5

**Review:**

**General Comments:**

The study presents an end-to-end hypothesis development, data collection, and testing that fits well the theme of the eye tracking workshop with clearly explained results. The authors propose a maze-solving paradigm while simplifying the eye movement complexity into fixations and saccades that match well the requirements of the task at hand. This to me, personally serves as one of the very few studies that present its reasoning for decisions made to define the model constraints. Except for few minor details that are shared in detailed comments, the study is well-designed and well-written. Those details if added by the authors will help other researchers to borrow a similar paradigm to replicate current results or expand to other domains using for instance eye + head tracking paradigms in VR. The reviewer kindly asks the authors to cover those minor missing information in order to improve the already high quality of the paper.

**Detailed Comments:**
- One general comment/suggestion: Eye position for the eye-tracking community is either the 2D position in an eye image or eye-part 3D coordinates in an eye-tracking headset. Therefore, I suggest the authors to use 2d gaze position or gaze on the screen instead.
- Please share the gaze accuracy and precision numbers for the users. This will provide an understanding for the reader to determine a possible contribution of the gaze tracking biases in the findings. Since the entire maze screen is 14 dVA it seems that each maze cell corresponds to 0.5-1 degree of visual angle (just by visually checking figure 1). Would this affect the results? Depending on how coarse vs. fine the eye movements scan the path, these numbers would possibly  (gaze accuracy vs. resolution) help the reader in generalizing the results for a different task on different screen sizes.
- Minor suggestion: While the authors mention that for each trial they perform a user-specific calibration routine. It is important to note that the eye tracking error by nature (usually) is a spatially variable measurement. Even though it won’t affect the strength of the results it would help to use a 3x3 homography to correct for gaze calibration errors across the screen, rather than subtracting a single vector from all gaze samples.
- Is it fair to say that after applying the low-pass filter of 4ms Gaussian window and then removing the gaze data of above 50 degrees/sec. We are left only with fixations and the saccades in between? Please clarify.
- Page 3 last sentence seem incomplete.
- Why does the EXIT model perform many saccades? Is it solely because the authors decided to add a 10 pixels per frame cap on its movement velocity? Or there are other reasons? One would expect the EXIT model to quickly jump to the exit if training converged perfectly.
- Please report the statistical tests performed on the un-normalized metrics as well.
- How would the model explain the predictive properties of the eye movements? This is shown in many studies especially for an active user, due to visual-motor delay, or task temporal constraints, the gaze follows some kind of predictive strategy. For instance in a ball-catching scenario when the target trajectory is a stereotype.
- For further expansion of this study, as we know human visual system benefits from a short post-saccadic suppression, would this approach be able to incorporate that if one wanted to use this to model precise properties of the eye movements in a less task-oriented paradigm but more eye-movement-loyal context. A paradigm where one would be able to model and replicate, higher frequency eye movements such as drift, microsaccades, tremor, etc. (of course give a new data collection system and design)

---

### Official Review · Reviewer_amcy · 2022-10-18
**Generative Ege Gaze Model**

**Rating:** 6
**Confidence:** 1

**Review:**

The paper proposes a generative model to generate eye gazes similar to the eye gaze of a human while solving a maze. The paper is well written and easily understandable. The methodology that is used is sound and the results are good too. The idea of not fitting a model directly to human data and instead simulating following an object in the maze made the generator explainable too.
However, my only doubt is that I still don't fully understand why it is important to have a generative gaze model, or how it can be generalized to other tasks,

---

### Meta-Review · Area_Chair_Zk53 · 2022-10-20

**Recommendation:** Accept (Oral)
**Confidence:** 5

**Metareview:**

The paper proposes a novel method to generative models of human eye movements. This is a very interesting domain that has picked up over the past few year and it has significant impact on understanding mental processes and furthermore computational goals that guide eye gaze. They propose a new architecture to train a generator for eye fixations and gaze shifts and they demonstrate that it can potentially generalize for multiple tasks. The authors also discuss limitations of current work highlighting the importance of further exploration (e.g. link of model fovea params to biological mechanisms, mental simulation). Per reviewer's recommendation this work can function as an example to other areas of research. The authors have done a very good job with the organization of the paper.

---

### Decision · Program_Chairs · 2022-10-20

Accept (Oral)